# Bacterial filamentation as a mechanism for cell-to-cell spread within an animal host

Tuan D. Tran [1], Munira Aman Ali[1], Davin Lee[1], Marie-Anne Félix[2] & Robert J. Luallen [1✉]

Intracellular pathogens are challenged with limited space and resources while replicating in a single host cell. Mechanisms for direct invasion of neighboring host cells have been discovered in cell culture, but we lack an understanding of how bacteria directly spread between host cells in vivo. Here, we describe the discovery of intracellular bacteria that use filamentation for spreading between the intestinal epithelial cells of a natural host, the rhabditid nematode *Oscheius tipulae*. The bacteria, which belong to the new species *Bordetella atropi*, can infect the nematodes following a fecal-oral route, and reduce host life span and fecundity. Filamentation requires UDP-glucose biosynthesis and sensing, a highly conserved pathway that is used by other bacteria to detect rich conditions and inhibit cell division. Our results indicate that *B. atropi* uses a pathway that normally regulates bacterial cell size to trigger filamentation inside host cells, thus facilitating cell-to-cell dissemination.

[1] Department of Biology, San Diego State University, San Diego, CA 92182, USA. [2] Institut de Biologie de l' École Normale Supérieure, Centre National de la Recherche Scientifique, INSERM, École Normale Supérieure, Paris Sciences et Lettres, Paris, France. ✉email: rluallen@sdsu.edu

Pathogenic invasion of host cells often provides microbes with a major benefit, including a high concentration of metabolic resources and protection from extracellular immunity[1]. An individual host cell, however, represents a small niche and it is advantageous for intracellular pathogens to spread to other cells in order to maximize the use of host resources. All intracellular microbes can spread via extracellular escape from the initially infected cell, a process that leaves them vulnerable to host defenses. Some microbes, however, have evolved unique mechanisms to invade uninfected host cells directly from an initially infected cell. To date, two main paradigms for cell-to-cell spreading by bacteria have been described, both of which converge on effector-driven manipulation of host actin. In one mechanism, *Listeria monocytogenes*, *Shigella flexneri*, and *Rickettsia spp.* can polymerize host actin for cytoplasmic propulsion of bacteria into the lateral membrane for uptake by neighboring cells[2]. *Burkholderia* spp. also uses actin-based propulsion, but the invasion of neighboring cells is facilitated by fusion of the lateral membranes[3]. In another mechanism, *Ehrlichia chaffeensis* can directly transfer from a phagocytic cell to a target cell through actin-dependent filopodia formation, although it is unclear if this mechanism is directly initiated by the pathogen[1,4]. Each of these bacterial mechanisms has only been directly observed in cell culture systems, raising the possibility of alternative mechanisms for cell-to-cell spreading and dissemination in vivo[5].

Bacteria often change their cell shape as a fitness strategy to survive or thrive in diverse environments. Filamentation is an example of this morphological plasticity, often used as a survival strategy in toxic or stressful environments. During filamentation, there is a dramatic increase in cell length as a bacterium divides longitudinally multiple times without daughter cell separation, often without septation[6,7]. Gram-negative bacteria are mainly known to undergo filamentation after exposure to environmental stressors, including DNA damage and β-lactam antibiotics, which induce the SOS response[6]. Several bacterial pathogens, such as *H. influenzae*, *P. aeruginosa*, *Y. pestis*, have been observed to filament in vivo, including inside host cells[8]. However, to our knowledge, filamentation has never been observed to be used for cell-to-cell spreading by any intracellular pathogen, including bacteria or fungi[9].

In this work, we describe the discovery of an intracellular bacterial pathogen that uses filamentation to directly spread from cell to cell in the intestine of an animal host. Furthermore, initiation of intracellular filamentation by this pathogen requires a highly conserved metabolic pathway used by divergent bacteria to detect rich conditions to delay divisome assembly.

## Results

### Discovery and characterization of *Bordetella atropi*. Wild-caught nematodes and their naturally associated pathogens have served as models to dissect different aspects of host–microbe interactions[10–12]. Given their transparency, these nematodes allow for the visualization and dissection of novel aspects of microbial pathogenesis and host immune/stress responses[13–15]. Known natural intracellular pathogens infecting free-living nematodes include viruses and microsporidia, but prior to now, no intracellular bacteria had been identified[16]. From ecological sampling, we isolated a wild *Oscheius tipulae* strain (JU1501) from rotting crab apples (Supplementary Fig. 1). *O. tipulae* is a rhabditid species that is commonly found in soil and decomposing vegetation[17]. In JU1501, coccobacilli-shaped microbes were observed to infect the intestinal epithelial cells (Fig. 1a). Since microsporidian parasites are the most common intracellular pathogens found in wild *C. elegans* and other free-living nematodes[18,19], this microbe was originally reported in a large survey of microsporidia based on morphology, despite the failure of microsporidian-specific 18 S primers to produce an amplicon[19]. We conducted fluorescence in situ hybridization (FISH) on mixed-stage populations of this *O. tipulae* strain and found that a mixture of microsporidian-specific 18 S probes failed to bind (Supplementary Fig. 2a). Given the lack of hybridization of this microbe to commonly infecting microsporidian species associated with wild nematodes, we suspected it may instead be a bacterium. Indeed, a universal probe to bacterial 16S rRNA showed strong hybridization with a multitude of coccobacilli inside the animals (Fig. 1b). In addition to these coccobacilli, we also observed a second bacterial morphology in some animals in the same sample in which long filaments extended throughout the intestine (Fig. 1c). Identical phenotypes were seen using a FISH probe specific to the 16S rRNA of this filamenting coccobacillus bacterium but not a nonspecific bacterial 16S probe (Supplementary Fig. 2b, c).

We isolated this bacterium on LB agar plates (strain name LUAb4) and verified that it could reinfect JU1501 *O. tipulae* animals that were sterilized of all horizontally transmitted microbes using a bleach treatment (as seen Fig. 1e–g). We sequenced the genome of LUAb4 and conducted a phylogenomic analysis to find that the bacterium represents a new species in a separate monophyletic clade of *Bordetella* (Fig. 1d), clustering with Bordetellae isolated from human respiratory specimens or the environment. We named this bacterium *Bordetella atropi*, after Atropos, the Greek Fate who cuts the thread of life (Supp. Text 1, Taxonomic Summary). Similar to other Bordetellae, *B. atropi* has the appearance of a coccobacillus morphology when in its non-filamentous form[20]. Infection by *B. atropi* has a severe effect on host fitness with an ~2.5-fold decrease in average lifespan and 90-fold decrease in fecundity (Fig. 1e–f). To determine the order of appearance of *B. atropi* phenotypes in vivo, we conducted a pulse-chase infection experiment and found that most animals had a majority of bacteria displaying either short or long intracellular filaments at 16 and 24 h post-infection (hpi), while most animals had a majority of bacteria in the coccobacilli form at 38 and 48 hpi (Fig. 1g). When we used confocal microscopy on FISH-stained animals at 34 hpi, we found that the average in vivo filament length of WT *B. atropi* was 51.9 μm with a maximum of 130 μm, or ~130 bacterial cell lengths (Fig. 1h).

### *Bordetella atropi* is an intestinal intracellular pathogen. Our observations of *B. atropi* infection by light microscopy suggested that coccobacilli were predominantly inside intestinal epithelial cells (Fig. 1a). To verify the intracellular nature and tissue tropism of *B. atropi*, we conducted transmission electron microscopy (TEM) of infected animals at 24 and 48 hpi. The intestine of *O. tipulae* is comprised of pairs of tubulating epithelial cells along the anterior-posterior axis, except at the most anterior portion of the intestine where four epithelial cells surround the lumen[21]. This intestine is separated from the body wall muscle, epidermis, and germline by a narrow space that forms the pseudocoelom (Fig. 2a). In uninfected animals, the intestinal cells can be distinguished by a relatively lighter electron density compared to other tissues, the presence of the intestinal lumen with microvilli, and uniform distribution of intracellular gut granules (Fig. 2b). *B. atropi* coccobacilli and filaments were distinguished from host cellular structures by a thin, wavy, electron-dense cell wall surrounding large, electron-poor nucleoids, similar to TEM images of intracellular *Bordetella bronchiseptica*[22]; these phenotypes were not observed in uninfected animals (Fig. 2c–e). At 24 hpi, we saw bacteria with this characteristic cell wall delineation inside intestinal cells, including coccobacilli and filaments that were

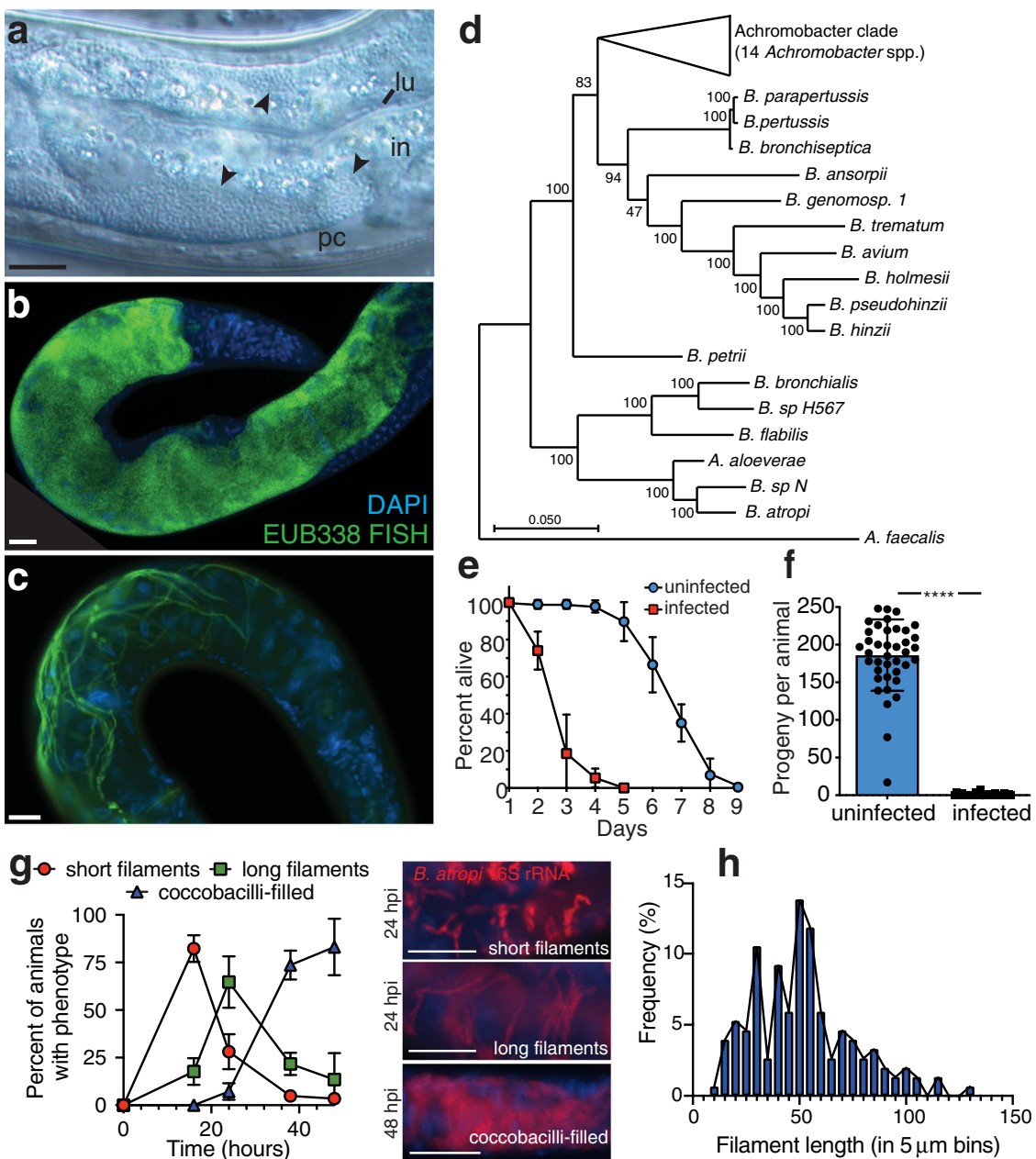

**Fig. 1 Discovery and characterization of *B. atropi* infection in *O. tipulae*. a** A multitude of coccobacilli (arrowheads) inside the intestinal cells (in) of *O. tipulae* strain JU1501. The intestinal lumen (lu) and pseudocoelom (pc) are indicated. **b, c** FISH using a universal bacterial 16S rRNA probe EUB338. Different animals shown from the same plate. Scale bars are 20 μm. **d** Phylogenomic tree of sequenced *Bordetella* and *Achromobacter* spp. with outgroup *Alcaligenes faecalis*. Branch lengths are the number of substitutions per site and branch points indicate percentage of trees with clustering of associated taxa. **e** Lifespan of JU1501 infected or uninfected with *B. atropi*, $n = 20$ animals in 3 independent experiments. Points represent mean values and error bars represent standard deviation (SD). **f** Broodsizes of JU1501, **** is $p < 0.0001$ by unpaired, two-tailed *t*-test with Welch's correction, $n = 20$ animals in 2 independent experiments. Error bars indicate SD and each point represents number of progeny from a single animal. **g** Pulse-chase infection time course, $n = 857$ animals binned over 2 independent experiments, error bars represent SD. Representative infection phenotypes are shown. Scale bars are 20 μm. **h** Histogram of in vivo filament lengths at 34 hpi.

unseptated with regularly spaced nucleoids (Fig. 2f, g). TEM only captures a portion of a filament length due to thin sectioning and the longest filament captured with this method was 10.06 μm (Fig. 2g), as compared to the average filament length of 51.86 μm as determined by tracing individual distinguishable filaments by confocal microscopy (see Fig. 1h). Of note, we also observed instances of putative nucleoid division or septum formation in the bacterial filaments (Fig. 2g). By 48 hpi, we see intestinal cells that are heavily infected with individual coccobacilli cells (Fig. 2h). Notably, in cases where we could see lumenal bacteria at both 24 hpi and 48 hpi time points, the lumenal coccobacilli bear the same morphological characteristics as those detected in the intracellular environment (see Fig. 2c, lower right for lumenal coccobacilli, and Fig. 2e). Using quantification, we found that 42% of images displayed intestinal filaments at 24 hpi ($n = 52$), and only 6.5% at 48 hpi ($n = 31$), showing a similar trend to the emergence of phenotypes in vivo via fluorescence microscopy (see Fig. 1g). Separately, TEM bolstered the intestinal tropism seen with *B. atropi* as 70.6% of images had intracellular intestinal infection ($n = 102$), but no infection was seen in non-intestinal

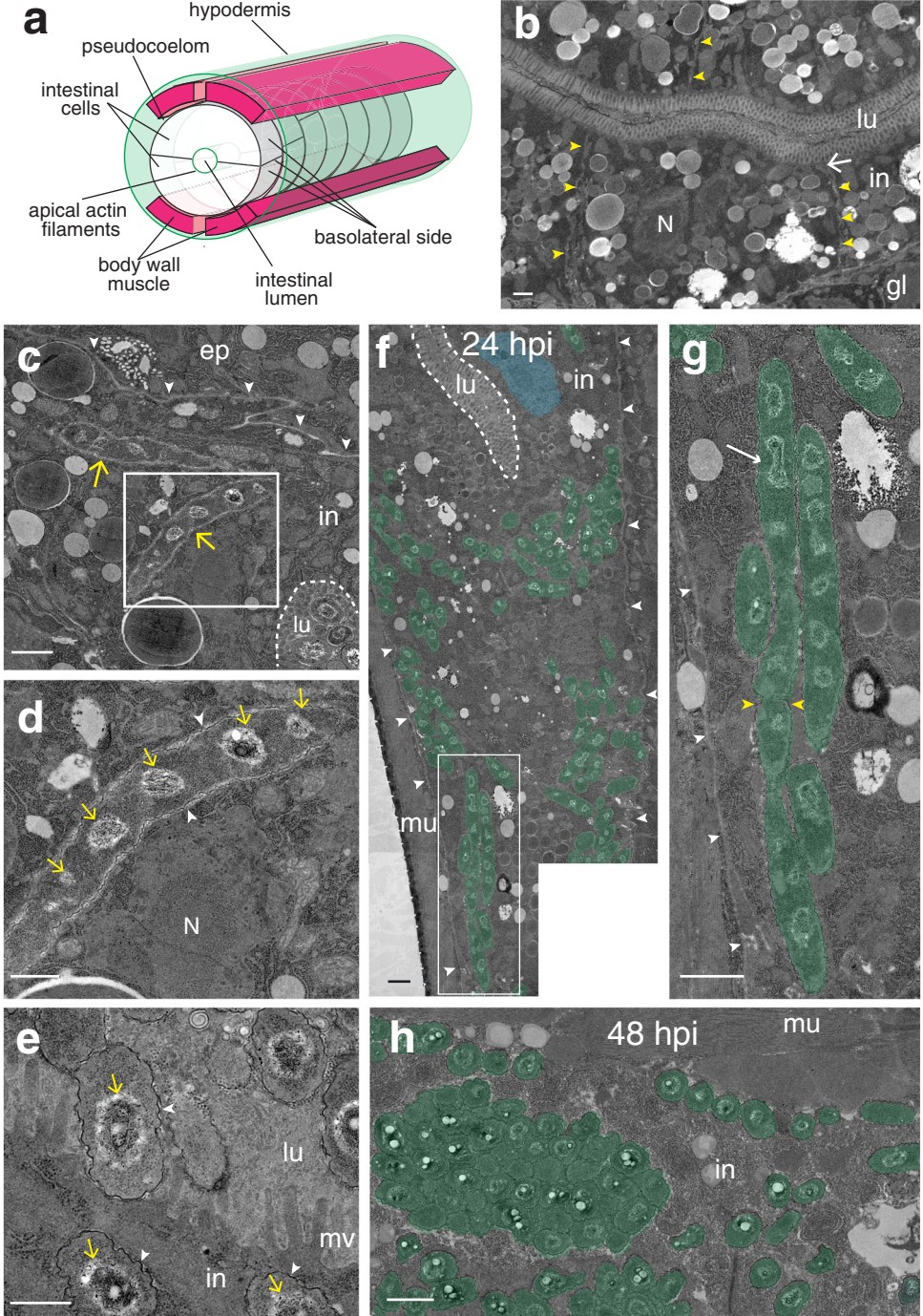

**Fig. 2 *B. atropi* infects and filaments inside intestinal cells of *O. tipulae*. a** A schematic of relative positions of intestine and surrounding tissues and compartments (not to scale, germline not shown for simplicity). **b** TEM of intestine of uninfected animals showing almost perpendicular cell-cell membranes delineation (yellow arrowheads), intestinal cell nucleus (N), and electron-dense apical junctions (white arrows). Intestinal cells (in), lumen (lu), and germline (gl) are indicated. **c** *B. atropi* filaments (yellow arrows) in intestinal cells at 24 hpi. The pseudocoelom (white arrowheads), intestinal lumen (dashed line), and epidermis (ep) are indicated. **d** Inset of white box in **c** with electron-dense cell wall (arrowheads) outlining the filament containing multiple nucleoids (yellow arrows). The intestinal nucleus (N) is indicated. **e** Coccobacilli in the intestinal cytoplasm (in) and lumen (lu) have the same electron-dense cell wall and large nucleoids as filaments. Microvilli (mv) are indicated. **f** A micrograph of an infected animal at 24 hpi with *B. atropi* coccobacilli and filaments (pseudocolored green) in the intestine with the intestinal nucleus (pseudocolored blue), pseudocoelom (arrowheads), and body wall muscle (mu) indicated. **g** Inset of white-boxed region in **f** with potential dividing nucleoids and septum indicated by arrow and yellow arrowheads, respectively. **h** A multitude of coccobacilli in infected intestine at 48 hpi. Scale bars are 1 μm in **b**, **c**, **f**, **g**, **h**, and 0.5 μm in **d** and **e**.

tissues, including the muscle ($n = 38$) and epidermis ($n = 73$). The lateral intestinal membranes appear disrupted in infected animals as they are often missing or at oblique angles relative to the intestinal lumen in infected cells. This is in contrast to uninfected cells where they are relatively perpendicular to the lumen (Supplementary Fig. 3). While the use of electron-dense delineation cannot unambiguously identify *B. atropi*, their absence in uninfected animals, the morphological similarities between those seen in intestinal cells and the lumen, as well as their high abundance at 48 hpi (similar to the FISH time course) indicate that these structures are characteristic of *B. atropi*. Together, these observations suggest that *B. atropi* is an intracellular pathogen that first filaments in intestinal epithelial cells and later septates into individual coccobacilli before exiting the host cell.

As a parallel approach to TEM, we also employed fluorescent dyes that mark the intracellular environment of intestinal cells and confocal microscopy to investigate the localization of *B. atropi* in infected animals. Using the dye carboxyfluorescein diacetate succinimidyl ester (CFSE), which strongly stains the cytoplasm of intestinal cells, we could observe the localization of *B. atropi* only within the boundary of the intestinal cell cytoplasm through z-stacks and orthogonal projections (Fig. 3a, b). Notably, we observed areas of CFSE-fluorescence clearing that overlap with the red fluorescence of *B. atropi* 16S probe (Fig. 3c, arrowheads in CFSE channel). This loss of CFSE fluorescence in these areas is likely due to bacterial filaments occupying these spaces, strongly supporting the cytoplasmic localization of *B. atropi* filaments (Fig. 3c).

Additionally, we used fluorophore-tagged phalloidin to stain host actin filaments, including apical ACT-5 and cortical actin filaments at the basolateral side of the intestine (Fig. 3d, e). In cases where phalloidin staining clearly showed the apical and basolateral perimeters of intestinal cells, we observed that *B. atropi* filaments and coccobacilli were localized within this actin staining. A representative line profile analysis shows that RFP-tagged bacteria lie in-between positions of basal and apical actin filaments in the intestine and close to nucleus, suggesting that *B. atropi* localizes within intestinal cells (Fig. 3f). We quantitated these images and found that 92.5% of infection events (i.e., coccobacilli or filaments) stay well within the boundaries of intestinal actin filaments, whereas 7.5% of infection events were ambiguous in their localization as there was direct overlap with the basal actin filament and we could not clearly assign localization to them (Fig. 3g). However, we did not observe any infection events that were clearly extracellular, where bacterial phenotypes appeared beyond the basal actin filament boundary of intestinal cells (0% extracellular), nor did we observe bacteria colocalizing with the body wall muscle. Separately, with phalloidin straining we also observed instances where bacterial filaments were pushing through the lateral actin filaments of intestinal cells, suggesting that these filaments were crossing through and disrupting the cell-cell boundary between intestinal cells (Fig. 3h, i), reminiscent of the TEM result (Supplementary Fig. 3). Altogether, these data strongly support the intracellular nature and intestinal tropism of *B. atropi*.

**Filamentation by *B. atropi* is used for cell-to-cell spread**. We hypothesized that filamentation by *B. atropi* is used as a mechanism for cell-to-cell spreading. Often, we observed cases where a single filament passed near multiple intestinal nuclei, indicating that the bacterium was simultaneously infecting neighboring intestinal cells (Fig. 4a). To test this hypothesis, we sought to isolate a non-filamenting mutant of *B. atropi*. Similar to *Bordetella avium*[23], our bacterium could be induced to filament

in vitro by switching growth from Luria broth (LB) to the richer medium Terrific broth (TB) (Supplementary Fig. 4a), allowing us to use in vitro selection to isolate filamentation mutants (Fig. 4b). We focused on one mutant, LUAb7, which formed less frequent and much shorter filaments in vitro compared to wild-type (Fig. 4c) but grew at the same rate (Fig. 4d). In vivo infection of *O. tipulae* found that LUAb7 also showed significantly decreased anterior-posterior spreading (Supplementary Fig. 4b, c). We confirmed that LUAb7 still infects the cytoplasm of intestinal cells in vivo using the cytoplasmic dye CFSE (Supplementary Fig. 5a, b). Further in vivo characterization of LUAb7 showed that the main phenotype of the bacteria in infected animals is coccobacilli throughout the course of infection, with areas filled with coccobacilli increasing as the infection progresses (Supplementary Fig. 5c, d). LUAb7 also forms much shorter filaments in vivo compared to WT, with an average length of 6.55 μm (Supplementary Fig. 5e). Even though LUAb7 failed to form long filaments in vivo, it still sufficiently reduced host genetic fitness to the same level as WT, suggesting that other virulence mechanisms may affect host fitness (Supplementary Fig. 5f, g).

The intestine of *O. tipulae* consists of left-right pairs of polarized epithelial cells running along the anterior-posterior (A-P) axis[21] (Fig. 2a). To quantify the cell-to-cell spreading capacity of WT and LUAb7, we pulse infected animals for a short period to limit the number of invasion events and then counted the number of intestinal nuclei that each contiguous infection event passes along the A-P axis. A contiguous infection event is assumed to be the result of replication from a single bacterium. For distinguishable filaments, we assume each filament is the replicative product from a single invasion event. However, we also observed infection foci, especially with LUAb7 infection. These foci are tightly packed clusters of coccobacilli, or a mixture of closely spaced short filaments (4–8 μm) and coccobacilli, and each focus is well separated from the other (Supplementary Fig. 6). We assume one infection focus is the result of one infection event in which the original invading bacterium underwent normal binary division; however, it is also possible that an infection focus is the result of more than one infection event. We observed both phenotypes in animals infected with either WT or LUAb7 strain, albeit at markedly different frequencies. For the WT strain, 95% of infection events were filaments and 5% were foci ($n = 162$), whereas 9% of infection events were distinguishable filaments (8–14 μm) and 91% were foci ($n = 131$) in case of LUAb7. When we followed single WT *B. atropi* filaments in vivo, they spanned an average of 5.7 and a maximum of 15 intestinal nuclei (Fig. 4e, f). By contrast, LUAb7 failed to consistently form long filaments in vivo, with replicating infection foci seen spanning an average of 1.9 and a maximum of 4 intestinal nuclei. Similar to *C. elegans*, intestinal nuclei in *O. tipulae* may undergo one nuclear division without cell division during development. Using intestinal membrane staining, we quantified an average of 1.9 intestinal nuclei per cell at both the anterior and posterior end (Supplementary Fig. 7). Therefore, WT *B. atropi* filaments spread laterally to an average of 3 cells and a maximum of 8 cells at 34 hpi, while the filamentation mutant remains largely restricted to a single cell but occasionally spreads to another cell. Altogether, these data indicate that *B. atropi* uses filamentation as an intracellular mechanism for cell-to-cell spreading.

**The UDP-glucose pathway controls *B. atropi* filamentation**. We identified the causative mutation in LUAb7 through whole-genome sequencing and single-nucleotide variations (SNVs) calling. The best candidate was a missense mutation found in *gtaB*, a UTP–glucose-1-phosphate uridylyltransferase. The mutation in *gtaB* led to an R17C change in a predicted catalytic

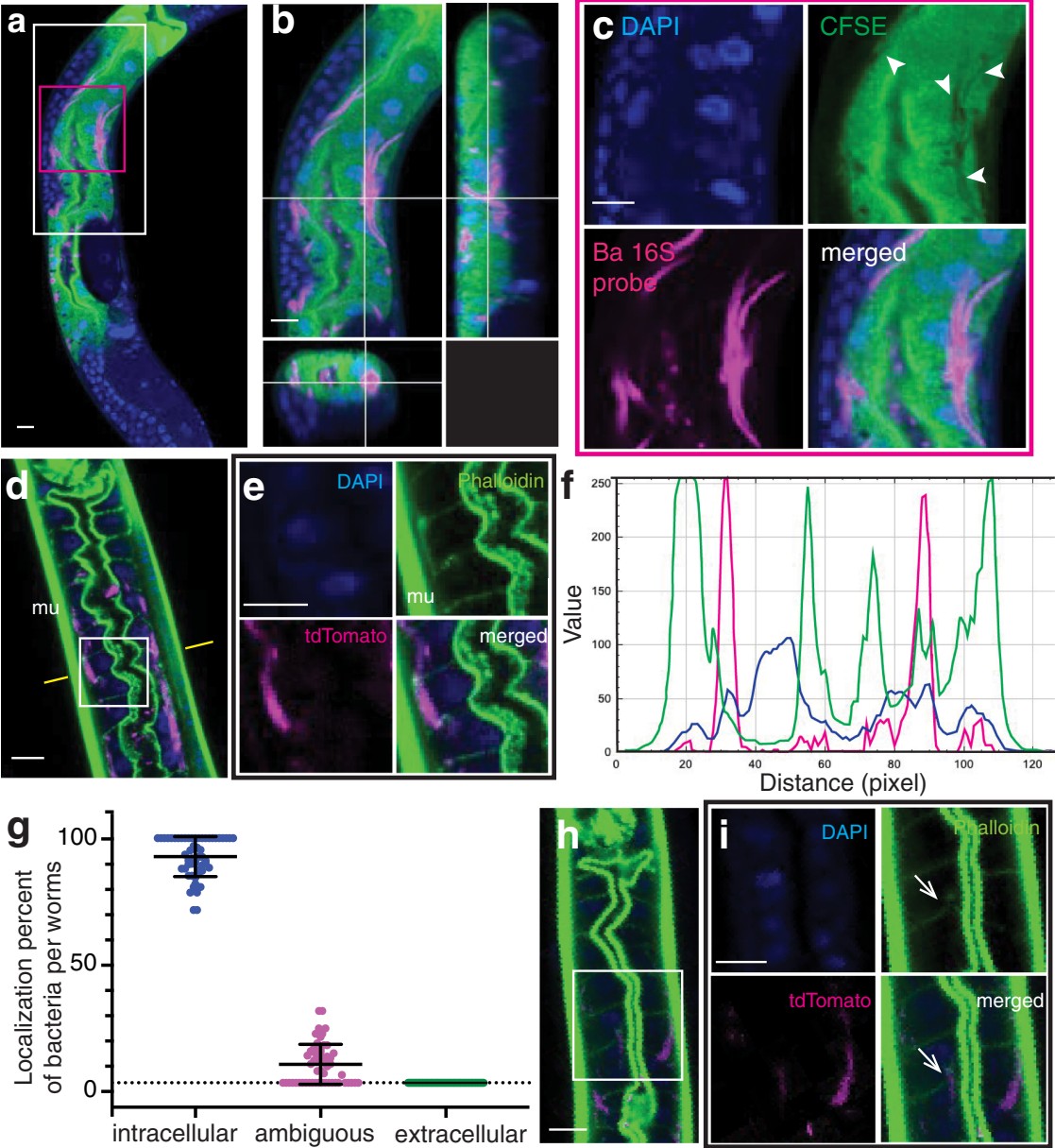

**Fig. 3 *B. atropi* filaments localize with intestinal intracellular staining. a** Representative confocal image of FISH-stained animals infected with *B. atropi* and stained with the cytoplasmic dye, CFSE, which strongly stains the intestinal cell cytoplasm. **b** Orthogonal view of white-boxed region in **a**. **c** Higher magnification of red-boxed region in **a** showing CFSE-cleared areas (white arrowheads) are overlapped by signals from 16S probe specific-FISH to *B. atropi*. **d** Representative image of animals stained with phalloidin. **e** Higher magnification of region indicated by white box in **d** showing *B. atropi* presence in intestinal cells outlined by basolateral and apical actin filaments (mu, body wall muscle is indicated). **f** Line profile analysis at the plane indicated by yellow lines in **d** showing localization of tdTomato (*B. atropi*) in-between Alexa488-phalloidin (actin). **g** Quantification of bacterial localization from the experiment conducted in **d**, with a total of $n = 60$ animals containing $N = 916$ infection events analyzed in 2 independent experiments. Means are shown and error bars represent SD. **h** A phalloidin-stained animal with a fragment of filament pushing through intestinal lateral actin. **i** Inset indicated in **h**. Scale bars are 10 μm.

arginine found in a highly conserved N-terminal motif (Fig. 5a)[24]. We complemented LUAb7 with *gtaB*[+] from *B. atropi* and found a rescue of in vitro filamentation (Fig. 5b) as well as in vivo filamentation, with similar levels of cell-to-cell spreading as wild-type (Fig. 4e, f). Additionally, deletion of *gtaB* from WT *B. atropi* resulted in a loss of filamentation in vitro, similar to *gtaB*[R17C] (Fig. 5d). Thus, *gtaB*[R17C] is the causative allele for loss of filamentation in LUAb7 and this single mutation abolishes cell-to-cell spreading of the pathogen.

GtaB (GalU in *E. coli*) catalyzes the conversion of glucose-1-phosphate to UDP-glucose, a substrate in the production of cell

wall components like osmoregulated periplasmic glucans (OPGs), lipopolysaccharide, and capsular polysaccharides (Fig. 4c). In *E. coli* and *Bacillus subtilis*, this UDP-glucose biosynthetic pathway serves as a metabolic sensor that links carbon availability to growth rate[25]. Excess UDP-glucose acts as a readout for rich conditions and temporarily inhibits FtsZ ring formation during replication for moderately larger progeny cells[26,27]. In *E. coli*, FtsZ ring inhibition is mediated by OpgH, a downstream glucosyl-transferase that is thought to sequester FtsZ monomers at the nascent division site in a UDP-glucose-dependent manner (Fig. 5c)[26]. We tested other members in this pathway for a role

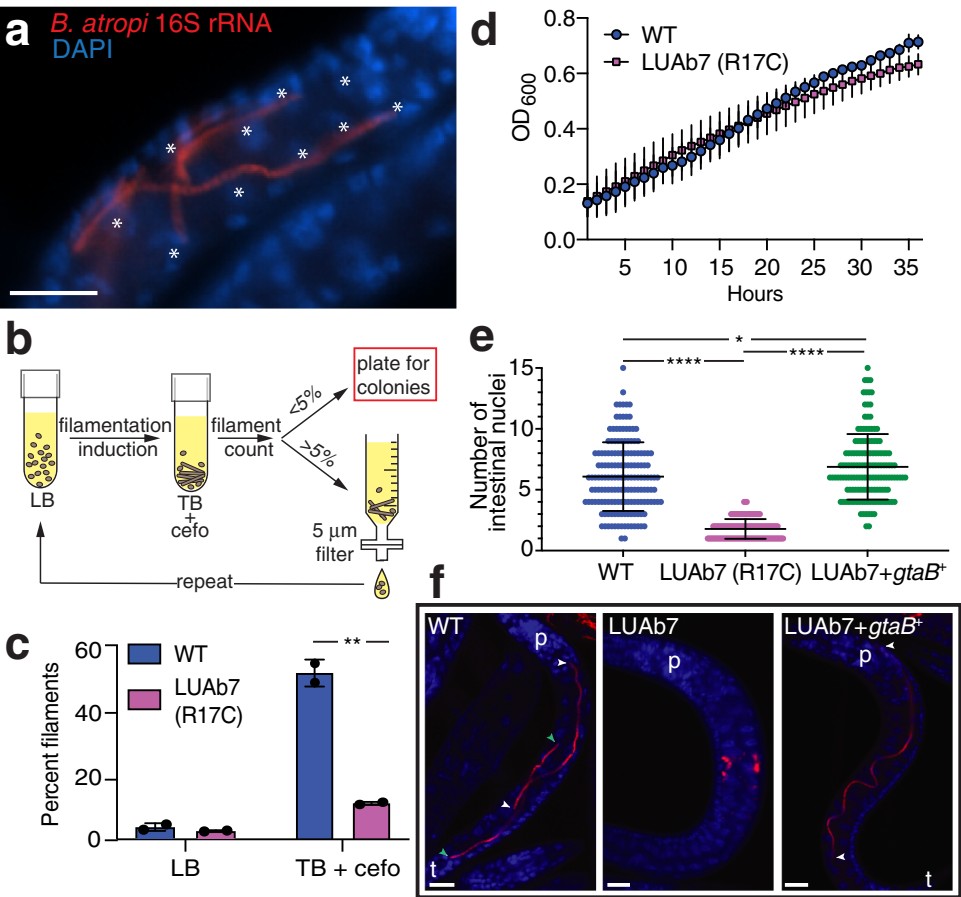

**Fig. 4 B. atropi filaments invade multiple intestinal cells. a** Filaments of *B. atropi* in a larval *O. tipulae* animal with host nuclei indicated by asterisks. **b** Schematic for selection of filamentation mutants. Filamentation was induced with TB + cefotaxime and counted for percent population with filaments, with >5% resulting in filtration followed by a repeat of induction and <5% resulting in plating to isolate single colonies. **c** Growth of *B. atropi* in LB or TB + cefotaxime, with percent filaments counted after 48 h at 32 °C. Bar graphs are presented with means and error bars represent SD. Each point represents an independent experiment, $p = 0.0054$ (**) by two-tailed $t$-test. **d** *B. atropi* strains grown in LB at 32 °C for 36 h with $OD_{600}$ taken every hour. **e** Quantification of *B. atropi* infection in vivo by the number of intestinal nuclei that a single filament or contiguous infection passes along on the anterior-posterior axis. Results are from $n = 161$ (WT), 129 (LUAb7), and 144 (LUAb7 + $gtaB^+$) infectious events from a total of 30 animals per group examined over 2 independent experiments, $p < 0.0001$ (****) or $p = 0.014$ (*) by Mann–Whitney two-tailed $t$-test. **f** Representative images from **e**, with filament ends (color-matched arrowheads), the pharynx (p) and tail (t) indicated. Scale bars are 10 μm.

in filamentation by knocking them out in *B. atropi*. Deletion of *opgH* resulted in a loss of filamentation in vitro, similar to LUAb7($gtaB^{R17C}$) (Fig. 5d). The *B. atropi* genome contained two copies of phosphoglucomutase (pgm) and deletion of *pgm2* but not *pgm1* resulted in loss of in vitro filamentation (Fig. 5d). Surprisingly, the knockout of *pgm1* did not phenocopy LUAb7 but instead led to an increase in vitro filamentation. This finding suggests that *pgm1* may negatively regulate UDP-glucose biosynthesis, as a similar role has been suggested for *galF* in *E. coli*, which may antagonize *galU* in the same pathway[28].

We then tested these knockouts in vivo. The *opgH* knockout phenocopied LUAb7($gtaB^{R17C}$), with reduced A-P spreading in the intestine (Fig. 5e). We verified that this decreased A-P spreading was not due to decreased infectivity, as LUAb7 ($gtaB^{R17C}$), Δ$gtaB$, and Δ$pgm1$ all had similar initial invasion events as WT at 16 hpi under our pulse-chase infection procedure (Supplementary Fig. 8). By contrast, the *pgm1* knockout showed no change in spreading in vivo compared to WT, consistent with in vitro data. While both the Δ$pgm2$ and Δ$gtaB$ strains showed an inability to the filament in vitro, we found that they were attenuated in vivo, as we saw no intracellular infection with these strains despite populations of >500 animals per replicate (Supplementary Fig. 9). In fact, we saw little to no bacteria in

the intestinal lumen with these knockouts, suggesting that a complete loss of some cell wall components may result in an inability to survive feeding by *O. tipulae*, perhaps leaving bacteria sensitive to the posterior grinder and/or destruction by extracellular defenses in the lumen. Altogether, these results support a requirement for the UDP-glucose biosynthetic pathway and *opgH* in inhibiting cell fission in *B. atropi*, but in a more extreme manner than seen in *E. coli* and *B. subtilis*, leading to long multinucleoid filaments rather than bacilli with modestly increased lengths. Overall, we propose a model where *B. atropi* detects the nutrient-rich intracellular environment of an intestinal cell through the UDP-glucose biosynthetic pathway and OpgH in order to initiate filamentation for cell-to-cell spreading (Fig. 5f).

## Discussion

Together, our findings describe bacterial filamentation as an in vivo mechanism for cell-to-cell spreading in a host epithelium. Several bacterial pathogens have been observed to the filament in vivo, including some that filament intracellularly, raising the possibility that other pathogens may use filamentation as a spreading mechanism[7,8]. For example, uropathogenic *E. coli* (UPEC) can form extremely long filaments inside umbrella cells, with conflicting data on whether these filaments are refractory to

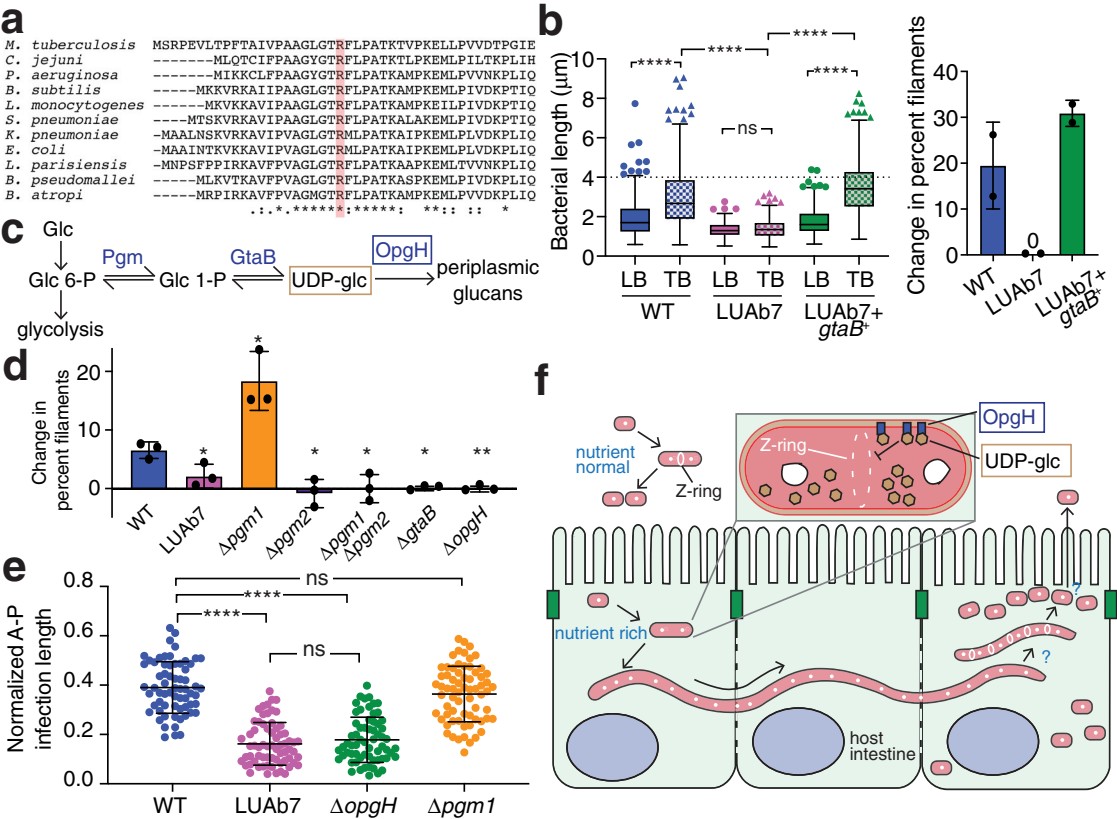

**Fig. 5 The UDP-glucose biosynthetic pathway is required for filamentation. a** CLUSTAL O multiple sequence alignment of GtaB from divergent bacteria with the predicted catalytic arginine highlighted. **b** In vitro filamentation of LUAb7($gtaB^{R17C}$) and the complemented LUAb7 + $gtaB^+$ as measured by bacterial length in LB and TB ($n = 150$, 2 independent experiments, Tukey's plot with boxes' lower and upper bounds representing 25th and 75th percentiles, respectively, center lines representing medians, lower whiskers representing minima and upper whiskers representing largest values less than 75th percentile plus 1.5 times interquartile range (IQR). Individual points are values that are larger than 75th percentile + 1.5 IQR. The cutoff for a filament is 4 μm (hatched line) (*left*), $p < 0.0001$ (****) or not significant (ns) by Kruskal-Wallis test followed by Dunn's post test with multiple-comparison correction. The change in percent filaments was measured (% filaments in TB − % filaments in LB) (*right*). Bar graphs are presented with means and SD error bars. Each point represents value from an independent experiment. **c** The UDP-glucose biosynthetic pathway converting glucose (*glc*) to UDP-glucose (*UDP-glc*). **d** *B. atropi* knockouts measured for the change in percent filaments. Results are from 3 independent experiments, $n > 400$ bacteria per replicate, $p < 0.05$(*) or $p = 0.0087$(**) by unpaired two-tailed *t*-test with Welch's correction compared to WT. Exact sample sizes for each group are provided in Source Data. Bar graphs show means with SD error bars. **e** Anterior to posterior (A-P) spr**e**ading of *B. atropi* mutants in vivo, normalized to animal length. Results are from $n = 65$ (WT), 67 (LUAb7), 64 (ΔopgH), and 68 (Δpgm1) animals examined over 2 independent experiments, $p < 0.0001$ (****) or not significant (ns) by Mann–Whitney unpaired two-tailed *t*-test. Error bars represent SD. **f** Model for *B. atropi* intracellular filamentation and spreading.

phagocytosis by macrophages after their release from host cells[29,30]. Some bacteria form intracellular filaments that can distort host cell shape in cell culture, as filaments of *Yersinia pestis* in macrophages can make protrusions in the host cell membrane[31] and filaments of *Salmonella enterica* serovar Typhimurium can grossly distort and elongate primary melanocytes[32]. We observed a similar phenotype in *B. atropi* whereby bacterial filaments push against and distort the lateral actin filaments of host intestinal cells. It is possible that exaggerated longitudinal growth is an in vivo strategy that intracellular pathogens can use to push through the lateral membranes in an epithelial layer for invasion of neighboring cells. For example, microsporidian parasites in *C. elegans* can form elongated intracellular stages (called meronts) that distort lateral intestinal membranes, leading to the formation of syncytial tissue and cell-cell spreading[13].

Despite *B. atropi* using filamentation for cell-to-cell spreading, we found that the LUAb7($gtaB^{R17C}$) mutant was still able to cause infection in *O. tipulae* without a noticeable change in host fitness. In fact, LUAb7 was able to invade intestinal cells at the same rate as WT *B. atropi*, suggesting that filamentation is not required for

host cell invasion (Supplementary Fig. 8). Since LUAb7 forms much shorter filaments compared to WT, we speculate that other virulence mechanisms may affect fitness downstream of invasion. For example, *B. atropi* infection may involve the production of bacterial toxins that have a severe effect on host fitness. Alternatively, non-filamenting coccobacilli may have faster in vivo replication and extracellular escape, allowing for quicker transmission to new animals and invasion of new intestinal cells within the same animal.

We found that filamentation induction by *B. atropi* is linked to nutrient availability through a highly conserved metabolic pathway that inhibits the bacterial divisome under rich conditions. These findings represent a new mechanism for the induction of filamentation in bacteria that is independent of the SOS response. Given the broad conservation of the UDP-glucose pathway and its role in delaying binary division on rich media[26,27], it is possible that the evolutionary steps for coopting this pathway for filamentation are minimal. For example, experimental over-expression of *opgH* in *E. coli* is sufficient to lead to filamentation, showing that altering the expression of a single gene in the pathway can induce this phenotype[26]. *B. avium* is a respiratory

pathogen of turkeys and was found to form filaments in TB similar to *B. atropi*, raising the possibility that filamentation in rich conditions is conserved among other clinically-relevant Bordetellae[23].

## Methods

**Nematode and bacterial strains**. Wild *O. tipulae* strain JU1501 was isolated from a rotting apple from Kerarmel, Plouezoc'h, France (GPS coordinates 48.65151, −3.85573) on August 6, 2008. *B. atropi* type strain LUAb4 was isolated from JU1501 through cleaning of dauer animals to remove contaminating microbes in M9 buffer + 0.25% SDS, 50 μg/mL of carbenicillin, 25 μg/mL of kanamycin, 12.5 μg/mL of tetracyclin, 100 μg/mL of gentamicin, 50 μg/mL of streptomycin, 37.5 μg/mL of chloramphenicol, and 200 μg/mL of cefotaxime[33]. After cleaning of JU1501, non-OP50-1 colonies were seen growing on nematode growth media (NGM) plates containing this strain. Single colonies were streaked to LB plates twice to ensure purity and the 16S rDNA was sequenced using primers 27 F (AGAGTTTGATCMTGGCTCAG) and 1492 R (GGTTACCTTGTTACGACTT).

**Fluorescent in situ hybridization (FISH)**. FISH was performed as described previously[34]. Briefly, experimental animals were harvested and washed 2–3 times with M9T buffer (M9 buffer, 0.05% TritonX-100), rinsed once with PBSTw buffer (1× PBS buffer, 0.1% Tween-20), and fixed with 4% paraformaldehyde in PBSTw for 45 min. Afterwards, fixed animals were pelleted by spinning at $1000 \times g$ for 1 min and washed four times with PBSTw before incubating with appropriate probes in hybridization buffer (900 mM NaCl, 20 mM Tris pH 7.5, 0.01% SDS) on a thermal shaker at 56 °C, 1000 rpm. After overnight incubation, animals were pelleted down, washed three times with FISH wash buffer (hybridization buffer, 5 mM EDTA) for 30 min each at 58 °C, 1000 rpm, then rinsed twice with PBSTw before mounting. Probes to the small subunit rRNA of bacteria and microsporidia were conjugated to either CAL Fluor Red 610 (CF610) or 5-Carboxyfluorescein (FAM). The probes used were MicroA-CF610 (CTCTG TCCATCCTCGGCAA), MicroC-CF610 (CAGAATCAACCTGGTGCCTT), and MicroE-CF610 (GTACTGGAAATTCCGTGTTC), *B. atropi*-specific 16S probe b004 (TTATC-CAGCGCCGTTTCTTTC), EUB338 (GCTGCCTCCCGTAGGAGT), and Alphaproteobacteria-specific 16S b002 (TGTACCGACCCTTAACGTTC).

**Fitness assays**. Lifespan assays were conducted using a bleached strain of JU1501 to remove all horizontally transmitted microbes. Assays were conducted in technical triplicate across two separate trials with $n = 20$ animals per plate seeded with either OP50-1 (the standard *E. coli* food source) or LUAb4 (*B. atropi*). Plates were grown at 23 °C and animals were scored every 24 h for death by touching non-motile animals with a pick. Accidentally killed and missing animals were censored from the count, but animals killed due to bagging were counted due to the possibility of infection-induced bagging. Brood size assays were conducted similarly, except $n = 20$ animals per condition were plated to individual plates across two separate trials. The number of viable (hatched) progeny was counted every 24 h and removed from the plate.

**Standard pulse infection procedure**. Two days prior to the infection, NGM plates were fully covered with 1 mL of an overnight culture of *B. atropi* strains or OP50-1 (OD$_{600}$ ~0.4–0.5) and bacteria were allowed to grow into a full lawn at room temperature. On the day of infection, animals JU1501 grown on standard OP50-1 food source were collected with M9T buffer, transferred onto *B. atropi* plates, and incubated 2 h at 23 °C to allow infection to occur. Then, animals were washed off plates into 15 mL centrifuge tubes, washed four times with M9T for 20 min, and plated onto the OP50-1 plates. Animals are incubated at 23 °C for indicated hours then processed by different assays and techniques as described below.

**Pulse-chase infection time course**. JU1501 animals were infected using the standard pulse infection procedure (see above). Animals were harvested at 16, 24, 36, and 48 hpi and FISH was performed as above with the *B. atropi* b004 probe. Images were taken and blinded for analysis, with animals binned as short filaments, long filaments, or filled with coccobacilli based on the majority phenotype seen in the animal. Uninfected animals or those with solely dispersed coccobacilli (representing recent invasion) were not counted. Samples were then unblinded and the percent of each phenotype at each timepoint was calculated separately for each replicate.

**Transmission electron microscopy (TEM)**. Animals were grown on OP50-1 to mixed stages were infected with wild-type *B. atropi* with the standard pulse infection procedure. At 24, 36, and 48 h post-infection, animals were harvested with M9 buffer, washed thoroughly with M9 for a couple times until the washing buffer looked sufficiently clear. Animals were then fixed with FGC fixative (2% paraformaldehyde, 2.5% glutaraldehyde in 0.1 M cacodylate buffer pH 7.4) for 2 h at room temperature on a rotator. Fixed samples were then kept in the same fixative at 4 °C until further processing for TEM. TEM was performed by the

Electron Microscopy Facility, Department of Cellular and Molecular Medicine at UCSD as described previously[34].

**Genome sequencing and assembly**. *B. atropi* strain LUAb4 was cultured from a single colony in LB and DNA was purified using the DNeasy Blood and Tissue Kit (Qiagen, 69504). Libraries were made using Illumina Nextera DNA Library from ~700 bp fragments and sequenced on Illumina MiSeq 2×250 bp by GeneWiz Inc. Unprocessed reads were assembled with SPAdes (v3.13.0) using the 'careful' parameter and testing 21, 33, 55, 77, 99, and 127 kmers[35]. Annotation was conducted with prokka (v1.14.0) with the parameters 'addgenes' 'genus Bordetella', and an evalue of 1e-06[36]. This Whole Genome Shotgun project has been deposited at DDBJ/ENA/GenBank under the accession JAHPZX000000000. The version described in this paper is version JAHPZX010000000.

**Phylogenomics analyses**. Phylogenomics was conducted using the UBGC pipeline as described[37], using 92 up-to-date bacterial core genes isolated from *B. atropi* and the following sequenced genomes: RHXM01000001.1 (*A. aegrifaciens*), UFQB01000001.1 (*A. agilis*), PYAL01000001.1 (*A. aloeverae*), AGUF01000105.1 (*A. arsenitoxydans*), CP013923.1 (*A. denitrificans*), VBUB01000010.1 (*A. dolens*), CP019325.1 (*A. insolitus*), GL982451.1 (*A. insuavis*), NJIG01000010.1 (*A. marplatensis*), QAYS01000001.1 (*A. mucicolens*), BCTK01000001.1 (*A. piechaudii*), OT017433.1 (*A. ruhlandii*), LGVG01000001.1 (*A. spanius*), LN831029.1 (*A. xylosoxidans*), CP013119.1 (*Alcaligenes faecalis*), FKIF01000001.1 (*B. ansorpii*), NC_010645.1 (*B. avium*), CP016170.1 (*B. bronchialis*), NC_019382.1 (*B. bronchiseptica*), CP016172.1 (*B. flabilis*), NEVR01000007.1 (*B. genomosp. 1*), NZ_CP012076.1 (*B. hinzii*), NZ_CP007494.1 (*B. holmesii*), NC_018828.1 (*B. parapertussis*), NC_002929.2 (*B. pertussis*), NC_010170.1 (*B. petrii*), NZ_CP016440.1 (*B. pesudohinzii*), and NZ_LT546645.1 (*B. trematum*). Alignments were made using UBCG (version 3.0) with parameter -a aa (using RAxML alignment based on amino acid sequences). Trees were made in MEGA X (version 10.0.5) using maximum likelihood with 500 bootstraps using AA substitution via JTT model and tree inference using Nearest-Neighbor-Interchange.

**Bacterial strains, plasmid construction, and gene knockout**. *B. atropi* expressing tdTomato was created by mini-Tn7 tagging as described before and the site of insertion was confirmed by arbitrary PCR and Sanger sequencing[38]. Primer sequences used for these strain constructions are listed in Supplementary Table 1.

The in-frame deletion was carried out as described previously[39]. Briefly, 1.5 kb fragments upstream and downstream of genes of interest (including the first and last 5–10 codons, respectively) were cloned and Gibson assembled together with the triple-digested vector pCVD443 using NEBuilder HiFi DNA Assembly Cloning Kit (New England BioLabs Inc., E5520S). Assembly mixture was then used to transformed *E. coli* SM10 λpir by electroporation (MicroPulser Electroporator, Bio-Rad). The parental *E. coli* SM10 strain was used to mate with wild-type *B. atropi*. Exconjugants were selected for the first recombination event with antibiotics and subsequently the second recombination event with sucrose. Putative knockout colonies were confirmed for deletion of the gene of interest by colony PCR with primers flanking the gene.

Complementation was performed by mating the filamentation mutant strain LUAb7 with *E. coli* SM10 that had been transformed with plasmid pANT4 constitutively expressing a wild-type copy of the gene *gtaB*.

**In vitro filamentation assay**. A single colony from a streaked plate was inoculated into an overnight culture for each bacterial strain. The overnight cultures (OD$_{600}$ ~0.4–0.5) were diluted to OD$_{600}$ ~0.2 with LB and 200 μL were used to seed 4.8 mL of either LB or TB media. Cultures were incubated at 32 °C, 100 rpm. After 2 days, all cultures were blinded. Each blinded culture was resuspended gently by pipetting slowly with P1000 pipette once to avoid breaking filaments and 7 μL of the culture was mounted on glass slides for imaging with a fluorescent Nikon Ni-E Eclipse. For each condition, several random frames from the slide were taken and a total of 500 microbes or more were counted. These microbes could be either filaments (longer than 4 μm), elongated cells (less than 4 μm but longer than a typical coccobacillus), or coccobacilli (~0.5–1 μm). Only the filamentous microbes were counted toward the filament percentage, calculated using the ratio of the number of filamentous microbes to the total number of cells.

**In vitro selection for filamentation mutants**. Wild-type *B. atropi* was subjected to filamentation induction by switching from LB to 5 mL of TB plus 200 μg/mL cefotaxime and grown at 32 °C, 100 rpm for 2 days. Cultures were syringe filtered through a 5.0 μm PVDF membrane (Millex SV) by gently, slowly pressing and 200 μL of the flowthrough was used to seed 4.8 mL of TB plus cefotaxime for another cycle of induction. Between each selection cycle, cultures were analyzed post-filamentation induction by gently resuspending with P1000 pipette once and mounting 3 × 7 μL to determine the filament percentage as described above. If greater than 5% of the population filamented, then the culture went through another round of selection. If less than 5% filamented, the flowthrough was plated on LB to isolate single colonies. Select colonies were tested for in vitro filamentation in TB, leading to the isolation of LUAb4.

**In vitro bacterial growth curve**. Overnight cultures of either WT or LUAb7 were diluted to $OD_{600}$ ~0.2 with LB. Each of the diluted cultures was added to 12 wells (total of 36) of a clear bottom, non-cell culture-treated 96-well plate. The peripheral wells were added with sterile DI water to minimize evaporation during $OD_{600}$ reading. $OD_{600}$ values were taken at 1 h interval for 36 h at 32 °C using VersaMax Tunable Microplate Reader (software SoftMax Pro 6.3) with settings for 30-s shaking before the actual reading of the $OD_{600}$. The final OD values at each time point for each condition were averaged from the 12 readings subtracted by the average blank reading (LB).

**Intestinal cell-to-cell spreading assay**. JU1501 was grown on OP50-1 food source and collected into a 15 mL centrifuge tube with M9T. Large animals (L3s, L4s, and adults) were allowed to settle to the bottom of the tube and the liquid on top containing mainly L1s and L2s was transferred to another 15 mL centrifuge tube. L1 and L2 animals were pelleted by centrifuging at $3000 \times g$ for 1 min and the animal pellets were divided into three parts and used for the standard pulse infection procedure as described above. After 2 h pulse infection, animals were washed with MTEG buffer (M9T buffer, 0.25 mM EDTA, 10 µg/mL gentamicin) to kill extracellular bacteria and plated on NGM plates seeded with OP50-1. At 34 hpi, animals were collected for FISH.

FISH samples were scanned from the top right corner to the left, bidirectionally, to look for animals with distinguishable infection events. Each contiguous infection event is expected to result from a single cellular invasion, with a bacterium either filamenting or undergoing binary division resulting in a cluster of replicating coccobacilli. A total of 15 animals were imaged per condition, each containing between 1–10 distinguishable infection events. Images were taken with Zeiss LSM 710 inverted confocal microscope (software Zen 2012 SP1 Black edition, Zeiss LSM 710 release version 8.1.0.0) in z-stacks and analyzed with Fiji image software (version 2.1.0/1.53 h) to count the number of whole intestinal nuclei that are fully passed by a single infection event along the A-P axis.

**Normalized anterior-posterior (A-P) infection length**. Animals were grown on OP50-1 food source and infected with standard pulse infection procedure (see above). At 34 hpi, animals fixed for FISH. Each condition was then imaged randomly for at least 30 animals with a fluorescent Nikon Ni-E Eclipse. Images were analyzed for normalized A-P infection length with Fiji image software (version 2.1.0/1.53 h). In brief, for each animal in a particular condition, the distance between the most anterior and the most posterior points of a distinct, contiguous infection area was measured. If an animal had more than one contiguous infection area, the distances were averaged. This averaged A-P length of contiguous infection areas was then normalized to the length of the same infected animal to give the normalized A-P infection length.

**Membrane dye and phalloidin staining of O. tipulae**. Animal fixation and cuticle reduction were adapted from[40]. Worms were fixed with pre-chilled 4% paraformaldehyde in PBS at room temperature on a rotator for 30 min and moved to 4 °C overnight. Animals were then washed three times with reduced cuticle buffer before incubating with fresh reduced cuticle buffer at 4 °C for 24 h. Afterwards, reduced animals were treated with 0.5 kU/mL collagenase type VII (Sigma–Aldrich, C0773) at 37 °C for ~32 h. Afterwards, animals were rinsed once with DI water, followed by 3 washes with PBS, and store in PBS at 4 °C for 1 day before staining. Animals were stained first with 0.5 mg/mL DAPI overnight. Excess DAPI was removed before staining with respective dyes. For membranes with a mixture of CellBrite Cytoplasmic membrane dye (Biotum, 30021) and BODIPY FL $C_5$-Ceramide (Invitrogen, B22650) at 400- and 1000-fold dilution, respectively, for 2 days. For phalloidin staining, Alexa 488-tagged phalloidin (Invitrogen, 400× stock, A12379) was added at 4× working concentration and incubated for 2 days. All the staining was carried out in a thermal shaker at 37 °C, 900 rpm.

**CFSE dye staining of O. tipulae**. Animals were infected with the pulse-chase procedure as described above. After 2 h of infection, animals were washed twice with MTEG buffer then wash once with M9T buffer. Animals were then pelleted down by spinning at $0.4 \times g$ for 30 s. 500 µL of CFSE staining buffer (M9, 2.5 mM serotonin, 40 µg/mL CFSE dye (Invitrogen, C1157)) was added and incubated for 3 h. Animals were then plated on 6 cm NGM plate seeded with OP50 and another 500 µL of CFSE staining buffer was laid on top of the plate until animals were collected for FISH at 34 hpi.

**Single-nucleotide variation (SNV) calling**. Strain LUAb7 was sequenced as described above using Illumina MiSeq. Reads were processed with cutadapt (v2.3) with parameters -u 10 -u -10 -U 10 -U 10 -m 50 to trim 10 bp from each end of every read and remove reads less than 50 bp[41]. Reads were mapped using bwa (v0.7.17-r1188) using the MarkDuplicates parameter[42]. SNVs were called using freebayes (v1.3.2-dirty) with parameter -C 50 requiring at least 50 supporting observations, a minimum mapping quality of 20, and base quality of 30[43], with the resulting data annotated by hand to look for SNVs with >90% of reads containing the variation. Raw reads from LUAb7 sequencing are available at Sequence Reads Archive (SRA) accession SRR14751734.

**Statistics and reproducibility**. All statistical analyses were performed with GraphPad Prism (version 9.1.2(225)). For all experiments where results are presented with qualitative representative images, each experiment was repeated two or three times independently with similar results.

**Taxonomic summary of Bordetella atropi**

Bordetella atropi *defines a new species*. This bacterium was originally found in its nematode host, *Oscheius tipulae* strain JU1501. It is orally transmitted, with no evidence of vertical transmission. The bacterium was cultured in pure culture in vitro in Luria Broth (LB) and other standard bacterial media and was amenable to conjugation with *E. coli* for transmission of plasmids. Phylogenomic analysis based on 92 highly conserved bacterial gene sequences placed this bacterium in the *Bordetella* genus as a new species.

*Life cycle and symptoms in its host*. Symptoms of infection were originally discovered by Nomarski light microscopy as small coccobacilli in the intestinal lumen and inside intestinal cells. The infection was passed from uninfected animals to infected animals through co-culture on the same plate, suggesting a fecal-oral route of transmission. Additionally, since the bacterium was isolated and cultured in vitro, animals were reinfected orally through exposure to the bacterium on a plate for as little as 2 h. Growth in LB resulted in mostly coccobacilli bacteria, with a small proportion of the population displaying flagellar motility. It is likely that the coccobacillus form of the bacterium is infectious to *O. tipulae*.

Observations of infection from fluorescent in situ hybridization (FISH) experiments using specific 16S rRNA probes and transmission electron microscopy (TEM) indicated that *B. atropi* has a filamentous morphology during early infection, which preceded the observation of hundreds to thousands of coccobacilli in late infection. The filamentous form was often observed to be simultaneously infecting intestinal cells and indicating invasion of neighboring cells. All infection phenotypes were only observed in intestinal cells, except for coccobacilli in the intestinal lumen. All post-embryonic stages showed signs of infection, including non-feeding dauer animals (if exposed prior to entering dauer).

The type strain (LUAb4) was isolated from a rotting apple below a wild apple tree near Kerarmel, Plouezoc'h, France (GPS coordinates 48.65151, −3.85573) on August 6, 2008.

The etymology of the type species name *B. atropi* is based on the filamentation phenotype (long threads) and the high level of pathogenicity to the cognate *O. tipulae* strain JU1501. This species was named after Atropos, the Greek Fate who cuts the thread of life.

**Reporting summary**. Further information on research design is available in the Nature Research Reporting Summary linked to this article.

## Data availability

This Whole Genome Shotgun project for B. atropi type strain LUAb4 has been deposited at DDBJ/ENA/GenBank under the accession JAHPZX000000000. The version described in this paper is version JAHPZX010000000. Raw reads from LUAb7 sequencing are available at Sequence Reads Archive (SRA) accession SRR14751734. Source data are provided with this paper.

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

## Acknowledgements

We would like to thank Prof. Aharon Oren for his expertise in prokaryotic nomenclature and assistance in naming *B. atropi*. Thank you to Dalaena Rivera for helping come up with the name *B. atropi*. We would like to thank Dr. Anca Segall and Dr. Nick Shikuma for providing bacterial plasmids, access to equipment, and assistance with bacterial genetics. Thank you to Dr. Aaron Reinke for editing the manuscript. California State University, CSUPERB New Investigator Award and San Diego State University (SDSU) University Grants Program (UGP) award to R.J.L. M.A.F. is an Investigator of the Centre National de la Recherche Scientifique, France. SDSU Rees-Stealy Research Fellowship (RSRF) and University Graduate Fellowship (UGF) award to T.D.T.

## Author contributions

T.D.T., M-A.F., R.J.L. conceptualized the project. T.D.T. and R.J.L. designed the experiments. T.D.T., M.A.A., D.L., and R.J.L. performed the experimental studies. T.D.T. and R.J.L. carried out data analysis and wrote of the manuscript.

## Competing interests

The authors declare no competing interests.
