## [Peer Review File · Nature Communications]

Bacterial filamentation as a mechanism for cell-to-cell spread within an animal hostREVIEWER COMMENTS

Reviewer #1 (Remarks to the Author):

The ms by Tran et al makes several novel observations and reduces them to mechanism in a satisfying way. It begins with the observations of filamentous intracellular bacteria in a nematode, demonstrating the localization to intestinal epithelia and that filaments extend beyond one cell and into neighboring cells. By sequencing the genome to define a new bacterial species, and selecting for mutants that fail to filament in vitro, they identified genes involved in filamentation. Spontaneous and engineered mutations in genes involved in regulation of septation resulted in filamentation mutants that failed to show long filaments in nematodes. The writing is clear and the data appear to be of high quality.

Reviewer #2 (Remarks to the Author):

In this study, Luallen and colleagues report on *Bordetella atropi* using filamentation as an intracellular mechanism of dissemination between intestinal epithelial cells of its host, the rhabditid nematode *Oscheius tipulae*. The authors show that filamentation requires a conserved nutrient-sensing pathway involved in the detection of rich conditions, leading to inhibition of bacterial division. The authors propose that *B. atropi* dissemination represents a novel mechanism of bacterial spread from cell to cell. In this reviewer's opinion, the impact of the study would be greatly improved if the authors could visualize instances where the formed filaments go across cell-cell contacts, thereby achieving cell-to-cell spread, as proposed by the authors.

1. To demonstrate the intracellular location of bacteria, the authors used confocal microscopy and showed overlap between intestinal cells (nuclei) and bacteria, but did not unambiguously demonstrate that the bacteria are intracellular. Moreover, given the low level of magnification used, the approach does not rule out the possibility that the bacteria may be located extracellularly, in between cells. One approach would be to visualize cell-cell contacts using a cytoskeleton marker (such as phalloidin) and use confocal microscopy to capture a full Z-scan from the apical to the basal sides of intestinal cells. The approach would not only demonstrate that the bacteria are indeed inside cells, but may also reveal how the filaments "cross" cell-cell contacts to penetrate neighboring cells. Any clue, even descriptive ones, would greatly improve the impact of the study.

2. In an effort to demonstrate the intracellular location of the bacteria, the authors backed up the interpretation of their confocal microscopy approach by using high-magnification electron microscopy. These results strongly suggest the presence of bacteria inside cells (Fig, 1h-j), but it is difficult to determine how the authors unambiguously identified the bacteria as well as the various host features such as in, lu, etc... Lower magnification images corresponding to the presented high-magnification zoom-in images would certainly help orient the reader here, and the results should be presented in the main figures, not the supplementary materials. As for unambiguously identifying the bacteria, the authors could use an antibody against a given bacterial cell surface structure, such as LPS, in conjunction with Protein A Gold.

3. The authors should characterize the LUAb7 mutant, as conducted in Fig. 1e-k. Does the mutant kill the host? Does it form infectious progeny in vivo?

4. In quantifying anterior-posterior spread as a measure of dissemination the authors assume that the numbers of initial invasion sites are similar for wild type bacteria and the LUAb7 mutant. The authors need to demonstrate that it is indeed the case. Is LUAb7 as invasive as wild type?

5. Given the magnification of the images presented in Fig. 2f, it is difficult to appreciate how the authors counted the number of intestinal nuclei that each contiguous infection event passes along the A-P axis. Representative, high-magnification images used for quantification should be presented, at least in the supplementary materials; and the authors should make it clear what exactly they refer to as "contiguous infection". Is that a mix of filaments or are they able to track individual filaments? If yes, why not measure filament length?

Reviewer #3 (Remarks to the Author):

The manuscript entitled: "Bacterial filamentation is an in vivo mechanism for cell to cell spreading" by Tran et al describes experiments with *B. atropis* infections of the nematode *Oscheius tipulae* that suggests that it forms filaments during infection and filamentation promotes cell-to-cell spread. The findings are intriguing, but the imaging is not sufficient to demonstrate intracellular localization and there are several important questions with respect to mechanism that remain to be answered. In addition, the language is often misleading and needs substantial editing. Major criticisms are enumerated below:

1. Regarding Figure 1a, What drives cluster formation? Does it occur in vitro prior to infection or only in vivo? Are the clusters uniform and what is their size (the image is too small in Fig. 1a. Is cluster formation a pre-requisite for filamentation? Are these IBCs?
2. The two different animals shown in Fig. 1b-c are so different from one another. Why? In c, green filaments are visible, in b, they are not. See #5, below for required quantification of images.
3. It isn't clear what is the rationale of this study, it is not articulated. The authors simply appear to be using FISH probes to "have a look". It is superficial and incomplete. The authors need to describe for the reader why (lines 67-68) it is significant that microsporidian-specific 18S probes failed to bind (Fig. S2a), while a universal probe to bacterial 16S rRNA showed strong hybridization with thousands of coccobacilli inside the animal. How do the authors conclude that there are thousands of cocci in Fig. 1b? It isn't clear. The images are too low-resolution for the authors' claims. How is the distinction made between short and long filaments and coccobacilli in Fig. 1g? Again, they do not have sufficient resolution for this distinction. Similarly, with their resolution, it is hard to say that the filaments in Fig. 2 are intracellular. Having a marked intestinal strain (such as *glp-4* in *C. elegans*) would be helpful, higher resolution, combined with intestinal dissection or membrane staining is essential. Some of this is shown in Fig. S3, this should be shown in the main manuscript, but more needs to be done. From these images, it is impossible to tell whether the bacteria are in the lumen or inside cells.
4. The TEMs are more convincing, especially Fig. 1j, although Figure 1i is difficult to interpret. How are these defined as coccobacilli?
5. In Fig. S5, the graph shows 20% filamentation, but the entire image above it shows only filaments. How was this determined? There is no quantification of the "representative images". In how many worms were filaments of X length observed, etc.
6. Why does the *pgm1* null strain have higher levels of filaments? How much (if any) is the UDP-glucose concentration reduced in the *pgm1* null compared to *pgm2* null?
7. How do the filaments (and not the cocci) spread from cell to cell, but the cocci (and not the filaments) leave the cell? How do cocci exit cells? The authors state that filamentation is linked to nutrient availability, but this does not explain the "defilamentation" observed at later times, when *B. atropis* is still inside a nutrient-rich intracellular environment. This remains to be explained.

Minor:

1. Line 32: this is misleading, because not all intracellular pathogens spread from cell to cell. This sentence should be modified.
2. Line 54-55: be precise as to which bacteria.
3. Line 65: what is meant by mixed populations of *O. tipulae*?
4. Line 71: of this bacterium is too vague.
5. Methods are in the supplemental data, which is annoying.

6. Don't refer to Fig. 3 before describing the rest of Fig. 2

We thank the reviewers for their time and helpful comments. Our responses are listed below in **blue**

Reviewer #1 (Remarks to the Author):

The ms by Tran et al makes several novel observations and reduces them to mechanism in a satisfying way. It begins with the observations of filamentous intracellular bacteria in a nematode, demonstrating the localization to intestinal epithelia and that filaments extend beyond one cell and into neighboring cells. By sequencing the genome to define a new bacterial species, and selecting for mutants that fail to filament in vitro, they identified genes involved in filamentation. Spontaneous and engineered mutations in genes involved in regulation of septation resulted in filamentation mutants that failed to show long filaments in nematodes. The writing is clear and the data appear to be of high quality.

We would like to thank you for your time reviewing our manuscript, and for your positive comments.

Reviewer #2 (Remarks to the Author):

In this study, Luallen and colleagues report on *Bordetella atropi* using filamentation as an intracellular mechanism of dissemination between intestinal epithelial cells of its host, the rhabditid nematode *Oscheius tipulae*. The authors show that filamentation requires a conserved nutrient-sensing pathway involved in the detection of rich conditions, leading to inhibition of bacterial division. The authors propose that *B. atropi* dissemination represents a novel mechanism of bacterial spread from cell to cell. In this reviewer's opinion, the impact of the study would be greatly improved if the authors could visualize instances where the formed filaments go across cell-cell contacts, thereby achieving cell-to-cell spread, as proposed by the authors.

1. To demonstrate the intracellular location of bacteria, the authors used confocal microscopy and showed overlap between intestinal cells (nuclei) and bacteria, but did not unambiguously demonstrate that the bacteria are intracellular. Moreover, given the low level of magnification used, the approach does not rule out the possibility that the bacteria may be located extracellularly, in between cells. One approach would be to visualize cell-cell contacts using a cytoskeleton marker (such as phalloidin) and use confocal microscopy to capture a full Z-scan from the apical to the basal sides of intestinal cells. The approach would not only demonstrate that the bacteria are indeed inside cells, but may also reveal how the filaments "cross" cell-cell contacts to penetrate neighboring cells. Any clue, even descriptive ones, would greatly improve the impact of the study.

- We appreciate this helpful feedback. We've added new data to address this comment as Fig. 3 in the manuscript. In Fig. 3a-c, we used CFSE to preferentially stain the intestine and conducted confocal microscopy. Orthogonal views of 3D reconstructions show that the *B. atropi* filaments are fully contained within the CFSE stain, with clearings in the cytoplasmic dye where red-stained *B. atropi* is located.

Separately, as recommended, we used fluorescent-labeled phalloidin to stain host actin in Fig. 3d-e. In these images, we quantified the percent of infection events of *B. atropi* that were intracellular, where apical actin (ACT-5) and basolateral actin were clearly present in the intestine. Finally, we also provided a clue to what might happen at the cell-cell contacts in Fig. 3h-i, and main text lines 184-187, as we see instances where actin at the lateral membrane is disrupted by filaments.

2. In an effort to demonstrate the intracellular location of the bacteria, the authors backed up the interpretation of their confocal microscopy approach by using high-magnification electron microscopy. These results strongly suggest the presence of bacteria inside cells (Fig, 1h-j), but it is difficult to determine how the authors unambiguously identified the bacteria as well as the various host features such as in, lu, etc... Lower magnification images corresponding to the presented high-magnification zoom-in images would certainly help orient the reader here, and the results should be presented in the main figures, not the supplementary materials. As for unambiguously identifying the bacteria, the authors could use an antibody against a given bacterial cell surface structure, such as LPS, in conjunction with Protein A Gold.

- We have separated the TEM data from Fig. 1 and added more context images in now Fig. 2, including lower magnification images. To help identify host structures, we have included a simplified diagram of *O. tipulae* body to help orient the reader (Fig. 2a). For example, the lumen (lu) of the intestine is identified the by presence of microvilli.

- We did not perform immunohistochemistry TEM to unambiguously identify the bacteria. There are several reasons to this. Firstly, it is generally difficult to perform immunohistochemistry in nematodes due to their tough cuticles. Secondly, although it is not unambiguous, we found specific bacterial phenotypes, including the presence of a wavy electron-dense cell wall surrounding electron-poor nucleoids, inside intestinal cells of infected animals at different time points but not in uninfected controls. We now explicitly indicate these phenotypes in Fig. 2d-e, and we have addressed this point in the main text (lines 151-155). Additionally, we see similar bacterial structures in the intestinal lumen and intestinal cytoplasm only in *B. atropi* infected samples (Fig. 2f-h). Lastly, we now have another line of evidence to support the intracellular nature of the bacteria, which is the absence of CFSE fluorescence in intracellular locations of the intestine wherever there are bacteria (filaments or coccobacilli) occupying those spaces (Fig. 3a-c, Supplementary Fig. 5a-b).

3. The authors should characterize the LUAb7 mutant, as conducted in Fig. 1e-k. Does the mutant kill the host? Does it form infectious progeny in vivo?

- The characterization of LUAb7 mutant, as conducted in previous Fig. 1e-k, is now provided as Supplementary Fig. 5c-g. We also addressed the questions of whether the mutant kill the host and whether it forms infectious progeny in vivo in the main text (lines 203-210).

4. In quantifying anterior-posterior spread as a measure of dissemination the authors assume that the numbers of initial invasion sites are similar for wild type bacteria and the LUAb7 mutant. The authors need to demonstrate that it is indeed the case. Is LUAb7 as invasive as wild type?

- We have provided new data as Supplementary Fig. 8 (main text, lines 271-274) to show that there is no difference in initial invasion by these bacterial mutants.

5. Given the magnification of the images presented in Fig. 2f, it is difficult to appreciate how the authors counted the number of intestinal nuclei that each contiguous infection event passes along the A-P axis. Representative, high-magnification images used for quantification should be presented, at least in the supplementary materials; and the authors should make it clear what exactly they refer to as "contiguous infection". Is that a mix of filaments or are they able to track individual filaments? If yes, why not measure filament length?

- Representative high-magnification images used for quantification in Fig. 4f (previous Fig. 2f) are provided in Supplementary Fig. 6. And, yes, the data in Fig. 4f represents the spreading of individual filaments in cases where we can track individual filaments. In cases where it is a mixture of closely spaced/overlapping short filaments with coccobacilli, we considered this as a single infection focus from one infection event to measure its spreading in the intestine in Fig. 4f, as we addressed this question in the main text (lines 215-224). Since the LUAb4 mutant is mostly found as infection foci (while WT are mostly filaments), we believe we are overestimating the spreading capacity of LUAb7 by measuring infection foci and assuming each focus is from a single invasion event.

Reviewer #3 (Remarks to the Author):

The manuscript entitled: "Bacterial filamentation is an in vivo mechanism for cell to cell spreading" by Tran et al describes experiments with *B. atrophi* infections of the nematode *Oscheius tipulae* that suggests that it forms filaments during infection and filamentation promotes cell-to-cell spread. The findings are intriguing, but the imaging is not sufficient to demonstrate intracellular localization and there are several important questions with respect to mechanism that remain to be answered. In addition, the language is often misleading and needs substantial editing. Major criticisms are enumerated below:

1. Regarding Figure 1a, What drives cluster formation? Does it occur in vitro prior to infection or only in vivo? Are the clusters uniform and what is their size (the image is too small in Fig. 1a). Is cluster formation a pre-requisite for filamentation? Are these IBCs?

- We apologize for any confusion the term "cluster" might have caused. In Fig. 1a, we did not use the term "cluster" to mean well-separated clumps of bacteria or imply they are inside host vesicles. Instead, we meant to convey that the intestinal cells were filled with tightly packed bacterial cells. We have changed the description of Fig. 1a to avoid this potential confusion.

- The questions about the "true" clusters and their characteristics and functions are of interest to us to investigate in future studies. Unfortunately, it is not in the scope of our current study.

2. The two different animals shown in Fig. 1b-c are so different from one another. Why? In c, green filaments are visible, in b, they are not. See #5, below for required quantification of images.

- The images in Fig. 1b-c were taken from our initial observation of this naturally-infected, wild-isolate of *O. tipulae*. Therefore, the images are from a mixed-stage, mixed infected population. We show two animals in Fig. 1b-c that are indeed different animals that are representative of two different morphologies of the bacteria that we observed in this naturally infected wild population. In Fig. 1b, an animal was filled with coccobacillus form of the bacteria with no filaments, whereas in Fig. 1c, bacterial filaments could be seen along the anterior-posterior axis of the animal. This is the initial observation that prompted us to investigate the distribution of these two phenotypes (and as we found out later, intermediate, or mixed phenotypes between these two extremes) during the course of infection in a controlled experiment (Fig. 1g).

3. It isn't clear what is the rationale of this study, it is not articulated. The authors simply appear to be using FISH probes to "have a look". It is superficial and incomplete. The authors need to describe for the reader why (lines 67-68) it is significant that microsporidian-specific 18S probes failed to bind (Fig. S2a), while a universal probe to bacterial 16S rRNA showed strong hybridization with thousands of coccobacilli inside the animal. How do the authors conclude that there are thousands of cocci in Fig. 1b? It isn't clear. The images are too low-resolution for the authors' claims. How is the distinction made between short and long filaments and coccobacilli in Fig. 1g? Again, they do not have sufficient resolution for this distinction. Similarly, with their resolution, it is hard to say that the filaments in Fig. 2 are intracellular. Having a marked intestinal strain (such as *glp-4* in *C. elegans*) would be helpful, higher resolution, combined with intestinal dissection or membrane staining is essential. Some of this is shown in Fig. S3, this should be shown in the main manuscript, but more needs to be done. From these images, it is impossible to tell whether the bacteria are in the lumen or inside cells.

- We would like to thank the reviewer for pointing out the lack of well-articulated rationale of our study. We are interested in understanding host-pathogen interactions using wild-caught nematodes and their naturally-associated pathogens as a model. When we discovered that strain JU1501 is indeed infected with potential intracellular bacterial pathogen, we reason that this system could be used to study direct in vivo aspects of intracellular bacterial infection. This forms the rationale for our study, which is now provided in the main text, lines 71-76.

- The importance of microsporidian-specific 18S probes failing to bind and the strong hybridization with universal bacterial 16S probe is provided in main text lines 81-84 and 87-89.

- "How do the authors conclude that there are thousands of cocci in Fig. 1b?" We have removed the word "thousands" as suggested to avoid the confusion it may cause due to the lack of resolution, and therefore quantitative power, in this image.

- "How is the distinction made between short and long filaments and coccobacilli in Fig. 1g?" We set a threshold of length larger than 4 μm as filaments (this is corresponding to the length of end-to-end 4 bacterial cells) whereas coccobacilli are 1 μm . Short filaments are within the range of 4-8 μm and long filaments are larger than 8 μm . For this figure, images were blinded for analysis to remove bias, with animals binned as short filaments, long filaments, or filled with coccobacilli based on the majority phenotype seen in the animal.

- With regards to the intracellular nature of the bacteria, we cannot make transgenic *O. tipulae* as easily as can be done in *C. elegans* using high copy arrays. Instead, we have fed the cytoplasmic dye CFSE to live animals to strongly stain the intestine and found that in infected animals, the presence of bacterial filaments and coccobacilli resulted in clearance of intestinal cytoplasmic fluorescence, strongly indicating that the bacteria are indeed intracellular. Additionally, we stained host actin with Phalloidin and quantified the percent of infection events of *B. atropi* that were intracellular as described above for Reviewer 2. This new data is in Fig. 3 of the manuscript.

4. The TEMs are more convincing, especially Fig. 1j, although Figure 1i is difficult to interpret. How are these defined as coccobacilli?

- We use the term coccobacilli based on the fact that Bordetellae are described as coccobacilli, which we have added to the manuscript (see lines 104-106). We have added a higher resolution image of a coccobacilli from TEM in Fig. 2e. Note that since *B. atropi* is actively replicating in the cytoplasm, some coccobacilli may appear longer due to normal increases of bacterial length during binary replication.

5. In Fig. S5, the graph shows 20% filamentation, but the entire image above it shows only filaments. How was this determined? There is no quantification of the “representative images”. In how many worms were filaments of X length observed, etc.

- We apologize for the confusion from the original image. We wanted to clearly show filaments leading us to magnify a small section of the image, and as a result, we lost the representative nature of the image. We have adjusted the image to show a mixture of filaments and some single and elongated cells as suggested. This data is in Supplementary Fig. 4 now instead of Supplementary Fig. 5.

- For the percentage of in vitro filamentation in Supplementary Fig. 4a (previously Supplementary Fig. 5), we took several images of each condition and directly measured the length of cells. Cells that are longer than 4 μm were binned as filament, typical coccobacillus length falls within 0.5-1 μm , and the intermediate length that is larger than 1 μm but less than or equal to 4 μm are considered as elongated cells. A total of 500 cells are counted and the ratio of filamentous cells (> 4 μm) over all cells (500) is the percentage of filaments. This is described in our method section (lines 439-451 in manuscript).

- To the question “In how many worms were filaments of X length observed?,” we are not sure which part of the figure the reviewer is referring to. Supplementary Fig. 4a is in vitro data. Supplementary Fig. 4b-c are in vivo data, but here we strictly measure the anterior-posterior spreading capacity of *B. atropi* and the LUAb7 mutant. We use this as a simple proxy for the spreading capacity of each bacterial strain and the quantification of representative images in Supplementary Fig. 4c was shown on Supplementary Fig. 4b lower panel. We did not measure filament length in these images. In the main manuscript, we did use another readout to quantify spreading capacity, as seen in Fig. 4f whereby we counted host intestinal nuclei as a proxy of cell-to-cell spreading, and we also quantified the percentage of each phenotype (filaments or infection foci) in either WT- or mutant LUAb7-infected animals (main text, lines 225-228).

6. Why does the *pgm1* null strain have higher levels of filaments? How much (if any) is the UDP-glc concentration reduced in the *pgm1* null compared to *pgm2* null?

- We found 2 putative homologs of *pgm* in the genome of our bacteria. The high levels of *in vitro* filaments for the *pgm1* knockout is indeed intriguing and we intend to investigate this further in a future study. We found that the overall phenotype of $\Delta pgm1$ *in vivo* appears to be similar to the WT when the anterior-posterior lengths of infection areas were measured (Fig. 5e). $\Delta pgm2$ has significantly attenuated phenotype and we think that *pgm2* may be the more canonical homolog, whereas *pgm1* might have evolved to perform a different function. *pgm1* could function as a negative regulator, reminiscent of the *galF* antagonizing *galU* in *E. coli*. We have addressed this potential in the main text (lines 265-269), and we would like to investigate this further in future studies.

- As mentioned above, the question of UDP-glucose levels in $\Delta pgm1$ and $\Delta pgm2$ is of interest to us to study subsequently. The answer to this question could give us some insight into the potential function(s) of *pgm1*, which, as stated above, we think may not perform a canonical function. But for now we believe this is out of the scope of the current study.

7. How do the filaments (and not the cocci) spread from cell to cell, but the cocci (and not the filaments) leave the cell? How do cocci exit cells? The authors state that filamentation is linked to nutrient availability, but this does not explain the “defilamentation” observed at later times, when *B. atropis* is still inside a nutrient-rich intracellular environment. This remains to be explained.

- We appreciate that the reviewer is interested in other stages of the bacteria in its infection cycle. These are outstanding questions of great interest to us that we would like to investigate in our future studies. Currently, we only have some clues to the first question as how filaments spread from cell to cell. We observed bacterial filaments pushing through the lateral actin filaments of host intestinal cells, suggesting physical distortion may be one of the mechanisms filamentation can spread cell-to-cell. This is mentioned in the discussion section in the main text, lines 301-305.

Minor:

1. Line 32: this is misleading, because not all intracellular pathogens spread from cell to cell. This sentence should be modified.

- We have modified as suggested, lines 34-36.

2. Line 54-55: be precise as to which bacteria.

- We have clarified as suggested, lines 59-60.

3. Line 65: what is meant by mixed populations of *O. tipulae*?

- We meant mixed-stage population, and we have clarified this in main text, line 85.

4. Line 71: of this bacterium is too vague.

- We have clarified this in the text, line 94.

5. Methods are in the supplemental data, which is annoying.

- We apologize for the inconvenience our original format caused. The methods are now added together with other sections in a single manuscript.

6. Don't refer to Fig. 3 before describing the rest of Fig. 2

- Fig. 2 and Fig. 3 are now Fig. 4 and Fig. 5, respectively. We understand that it may be inconvenient for the readers to have to refer to a panel from the previous figure.

However, the quantification of the rescued strain was done in the same experiment with WT and the mutant LUAb7, and for this reason we feel that they should belong to the same panel in Fig. 4.

REVIEWERS' COMMENTS

Reviewer #2 (Remarks to the Author):

The authors did a good job at addressing my comments. I have no further request, this is a great story.

Reviewer #3 (Remarks to the Author):

In this revised manuscript, the authors have done a good job of addressing previous criticisms. The addition of phalloidin staining is a nice addition and adds clarity. I have minor comments.

Fig. 1g is not visible. It is too small to detect long filaments. Enlarge.

Line 61 Define CFSE

Lines 300 typhimurium should be capitalized and not italicized

The authors need to emphasize that when *B. atrophi* is not able to filament, it still causes infection. So filamentation is one mechanism, but not the only one. How does the infection proceed when filamentation is not possible? Although understanding cocci transmission is beyond the scope of this study, the authors should speculate in the Discussion and re-emphasize this point.

Line 712 Fig. 2 legend should be white arrowheads (not yellow)

Line 737 animals should not be plural

Please change "bringing up the possibility" with raising the possibility throughout the ms.

Thank you to the reviewers for their helpful comments. You've all helped made the manuscript better! Our responses to the most recent comments are listed below in **blue**

Reviewer #2 (Remarks to the Author):

The authors did a good job at addressing my comments. I have no further request, this is a great story. **Thank you!**

Reviewer #3 (Remarks to the Author):

In this revised manuscript, the authors have done a good job of addressing previous criticisms. The addition of phalloidin staining is a nice addition and adds clarity. I have minor comments. **Thank you!**

Fig. 1g is not visible. It is too small to detect long filaments. Enlarge. - **Enlarged as requested.**

Line 61 Define CFSE - **Changed as requested.**

Lines 300 typhimurium should be capitalized and not italicized - **Changed as requested.**

The authors need to emphasize that when *B. atrophi* is not able to filament, it still causes infection. So filamentation is one mechanism, but not the only one. How does the infection proceed when filamentation is not possible? Although understanding cocci transmission is beyond the scope of this study, the authors should speculate in the Discussion and re-emphasize this point. – **We've added a paragraph to the Discussion to address this.**

Line 712 Fig. 2 legend should be white arrowheads (not yellow) - **Changed as requested.**

Line 737 animals should not be plural – **For this, we believe 'animals' should be plural as more than one animal was measured. We changed the context of the figure legend to make it clear that we measured multiple animals, similar to the animal shown in panel d (of Figure 3).**

Please change “bringing up the possibility” with raising the possibility throughout the ms. - **Changed as requested.**